# CT in the Differentiation of Gliomas from Brain Metastases: The Radiomics Analysis of the Peritumoral Zone

**DOI:** 10.3390/brainsci12010109

**Published:** 2022-01-14

**Authors:** Lucian Mărginean, Paul Andrei Ștefan, Andrei Lebovici, Iulian Opincariu, Csaba Csutak, Roxana Adelina Lupean, Paul Alexandru Coroian, Bogdan Andrei Suciu

**Affiliations:** 1Radiology and Medical Imaging, Clinical Sciences Department, “George Emil Palade” University of Medicine, Pharmacy, Science, and Technology, 540139 Targu Mures, Romania; go2lucian@yahoo.com; 2Interventional Radiology Department, Târgu Mureș County Emergency Clinical Hospital, 540136 Targu Mures, Romania; 3Department of Biomedical Imaging and Image-Guided Therapy, General Hospital of Vienna (AKH), Medical University of Vienna, 1090 Vienna, Austria; 4Anatomy and Embriology, Morphological Sciences Department, “Iuliu Hațieganu” University of Medicine and Pharmacy, 400012 Cluj-Napoca, Romania; iulian_opincariu@hotmail.com; 5Radiology and Imaging Department, Cluj County Emergency Clinical Hospital, 400006 Cluj-Napoca, Romania; andrei1079@yahoo.com (A.L.); csutakcsaba@yahoo.com (C.C.); paulcoroian@yahoo.com (P.A.C.); 6Radiology, Surgical Specialties Department, “Iuliu Hațieganu” University of Medicine and Pharmacy, 400006 Cluj-Napoca, Romania; 7Histology, Morphological Sciences Department, “Iuliu Hațieganu” University of Medicine and Pharmacy, 400012 Cluj-Napoca, Romania; roxanalupean92@gmail.com; 8Obstetrics and Gynecology Clinic “Dominic Stanca”, Cluj County Emergency Clinical Hospital, 400006 Cluj-Napoca, Romania; 9The First Surgical Clinic, Târgu Mureș County Emergency Clinical Hospital, 540136 Targu Mures, Romania; suciubogdanandrei@yahoo.com; 10Anatomy, Morphological Sciences Department, “George Emil Palade” University of Medicine, Pharmacy, Science, and Technology, 540139 Targu Mures, Romania

**Keywords:** brain metastases, computer-aided diagnosis, computed tomography, glioma, glioblastoma, radiomics, texture analysis

## Abstract

Due to their similar imaging features, high-grade gliomas (HGGs) and solitary brain metastases (BMs) can be easily misclassified. The peritumoral zone (PZ) of HGGs develops neoplastic cell infiltration, while in BMs the PZ contains pure vasogenic edema. As the two PZs cannot be differentiated macroscopically, this study investigated whether computed tomography (CT)-based texture analysis (TA) of the PZ can reflect the histological difference between the two entities. Thirty-six patients with solitary brain tumors (HGGs, *n* = 17; BMs, *n* = 19) that underwent CT examinations were retrospectively included in this pilot study. TA of the PZ was analyzed using dedicated software (MaZda version 5). Univariate, multivariate, and receiver operating characteristics analyses were used to identify the best-suited parameters for distinguishing between the two groups. Seven texture parameters were able to differentiate between HGGs and BMs with variable sensitivity (56.67–96.67%) and specificity (69.23–100%) rates. Their combined ability successfully identified HGGs with 77.9–99.2% sensitivity and 75.3–100% specificity. In conclusion, the CT-based TA can be a useful tool for differentiating between primary and secondary malignancies. The TA features indicate a more heterogenous content of the HGGs’ PZ, possibly due to the local infiltration of neoplastic cells.

## 1. Introduction

The differentiation between high-grade gliomas (HGGs) and solitary brain metastases (BMs) is crucial, as they imply separate clinical and surgical management strategies [1], as well as different clinical outcome and overall survival rates [2,3]. The imaging characteristics of the two entities often overlap, as they both may present with a necrotic center, surrounding edema, and variable enhancing margins [4]. These common imaging features can lead to misclassification in almost half of their encounters [5].

The peritumoral zone (PZ) of the two entities has specific microscopic characteristics. Radiologically, the PZ represents the brain area surrounding the tumor, which exhibits no contrast enhancement. This area is of several centimeters in width around the tumor and is the site of specific cellular, molecular, and radiological alterations [6]. Since HGGs (frequently glioblastomas) tend to invade the surrounding structures, the adjacent white matter exhibits neoplastic cell infiltrates [4,7], which may be located several centimeters beyond the contrast-enhancing lesion [8]. On the other hand, the PZ of BMs is believed to be comprised of only pure vasogenic edema since secondary malignancies tend to displace rather than invade surrounding tissues [9,10]. These microscopic characteristics are expected to produce alterations of the PZ density on computed tomography (CT) images, but such changes are too subtle to be visible to human perception. 

In the last decade, the field of computer-based applications that aim to augment medical imaging diagnosis has grown exponentially. One such computer-based application is radiomics, which is a quantitative approach to medical imaging, that through advanced mathematical analysis, tries to improve the imaging-interpretation process [11]. The basic principle of radiomics relies on the presumption that medical images reflect “microscopic” disease-specific processes that cannot be assessed by the human eye, therefore not being accessible through the traditional visual inspection [11,12]. Texture analysis (TA) is a radiomics subfield that represents a method for extracting and processing specific parameters that quantify pixel intensity and variation patterns. The TA parameters offer a quantitative and comprehensive representation of image content [13,14] and can reflect information about the tissue microenvironment. Texture parameters’ role in imaging diagnosis has been extensively researched, especially for oncologic diseases, where these parameters have been demonstrated to be associated with histopathologic correlates (such as tumor grade and cellular processes), genetic features, and even clinical outcomes and prognosis [15]. So far, most TA studies involving gliomas have focused on solid tumor components [16,17], while the PZ remained relatively unexplored.

To the best of our knowledge, this is the first study that aimed to extract texture information from the PZ of solitary brain tumors based on CT images. Our objective was to investigate whether the resulting parameters may help in the non-invasive distinction of HGGs and BMs, and therefore aid in the diagnosis of the newly encountered solitary brain tumors.

## 2. Materials and Methods

### 2.1. Patients

This Health Insurance Portability and Accountability Act-compliant, single-institution, retrospective pilot study was approved by the institutional review board, and informed consent was waived due to the study’s retrospective nature. In our radiology database, a keyword search using the terms “brain + tumor”, “glioma”, “glioblastoma” and “brain + metastases” was conducted to identify patients who underwent contrast-enhanced CT (CECT) scans between May 2017 and March 2020. The original search yielded 686 reports. Each report was analyzed by one researcher who excluded all reports referring to extra-axial (*n* = 54), infratentorial (*n* = 60), intraventricular (*n* = 5), and multiple intra-axial tumors (*n* = 68). The medical records of the remaining 499 patients were retrieved from the archive of our healthcare unit and investigated for disease-related data. Further exclusion criteria were the benign, infectious or inflammatory nature of the lesion (*n* = 58), circumscript or low-grade gliomas (*n* = 29), the absence of a final histopathological result (*n* = 35), and examinations of recurrences of malignant lesions (*n* = 54). Furthermore, all remaining studies were reviewed by one radiologist who excluded non-enhanced CT scans (*n* = 71) and post-operatory control studies (*n* = 186), as well as investigations where the tumor was affected by artifacts (*n* = 9) and all lesions in which the PZ measured less than 10 mm (*n* = 21). 

### 2.2. Reference Standard

The final population comprised 17 patients with HGGs and 19 subjects with BMs. Gliomas were classified according to the 2016 World Health Organization (WHO) classification: glioblastomas, *n* = 11; anaplastic astrocytoma, *n* = 4; anaplastic oligodendroglioma, *n* = 3; anaplastic oligodendroglioma, *n* = 1. The primary tumors involved in the development of BMs included: pulmonary cancers, *n* = 6; melanomas, *n* = 5; breast cancers, *n* = 4; renal carcinoma, *n* = 2; colorectal carcinoma, *n* = 2. All lesions underwent histopathological analysis following their partial or complete surgical removal.

### 2.3. Image Acquisition and Interpretation

All examinations were performed on the same unit, Siemens Somatom Sensation, 16 slices (Siemens Medical Solutions, Forchheim, Germany), using a standard imaging protocol. The parameters of the CT scan were 120 kV, 200 mAs, pitch value 0.8, collimation 128 × 0.6 mm, and slice thickness 0.75 mm. The reconstruction algorithm was slice thickness of 3 mm in the axial plane and 2 mm in the coronal and sagittal planes, spacing 3 mm, and a window width and level of 2500/500 for soft tissue and 350/20 for bone. CECT scans were obtained following the injection of 80–140 mL of nonionic iodinated contrast material at a concentration of 350 mg/mL (iohexol (Omnipaque 350; Daiichi-Sankyo Health Care, Tokyo, Japan)) at a rate of 2–3 mL/s.

On a dedicated workstation (Advantage workstation 4.7 edition, General Electric, Boston, MA, USA), all examinations were reviewed by one radiologist, blinded to the final diagnosis. On each axial CECT, the slice considered to be the most representative for the peritumoral region was chosen to include both the tumor borders and as much as possible of the adjacent edema. The peritumoral region was defined as the area between the solid enhancing part of the tumor and the margins of the hypodense zone that surrounded the lesion.

### 2.4. Texture Analysis

The traditional approach of radiomics consists of four steps: image segmentation using regions of interest, feature extraction, feature selection, and prediction.

#### 2.4.1. Image Pre-Processing and Segmentation

The previously selected axial CECT slices were retrieved in Digital Imaging and Communications in Medicine (DICOM) format and further imported into texture analysis software, MaZda version 5 (Institute of Electronics, Technical University of Lodz, Lodz, Poland). A two-dimensional region of interest (ROI) was defined by placing a seed in the approximate center of the peritumoral edema, and the software automatically delineated the zone based on gradient and geometric coordinates. There were no restrictions regarding the ROIs’ width because the neoplastic cell infiltration could have been located beyond 1 cm from the contrast-enhancing border and up to several centimeters deep [6,8]. When necessary, manual adjustments were performed (Figure 1).

#### 2.4.2. Feature Extraction

Before extracting the texture features, each ROI was normalized by using a limitation of dynamics to *μ* ± 3*σ* (*μ* = gray-level mean; *σ* = gray-level standard deviation) to counteract the contrast and brightness variations that can affect the true texture of the image. The texture features (parameters) were automatically extracted by the software after the definition and positioning of every ROI. From each PZ, a total of 275 parameters were computed [18]. The major parameter classes as well as the computation settings are displayed in Table 1.

#### 2.4.3. Feature Selection

To identify which were the best-suited texture parameters for differentiating between the two histopathological groups, two steps were applied successively. The first step comprised using two feature reduction methods based on probability of classification error and average correlation coefficients (POE + ACC) and Fisher coefficients (F, the ratio of between-class to within-class variance). Each method provided a set of ten texture features that were previously described as having the best ability to discriminate between classes [19].

Second, the absolute values of the selected parameters were compared between the two groups by computing the Mann–Whitney U test (univariate analysis). The statistically significant level was set at a p-value of below 0.0023 after Bonferroni correction (which implied dividing the classic 0.05 level by 19, of which 17 represented the unique parameters that resulted after applying the reduction techniques, plus age and sex). All texture parameters that showed univariate analysis results above this threshold were excluded from further processing.

#### 2.4.4. Class Prediction

In the first step, the parameters that showed statistically significant results in the univariate analysis underwent receiver operating characteristics (ROC) analysis. The DeLong et al. technique was used to compute the ROC curves, and the binomial exact confidence intervals (CI) for the areas under the curve (AUC) were stated. The optimal cutoff values for predicting primary malignancies were determined using a common optimization step that maximized the Youden index. Specificity (Sp) and sensitivity (Se) were calculated from the same data, without other adjustments, using a 95% CI.

In the second step, a multiple regression (multivariate analysis) was conducted to investigate which texture features could independently predict the presence of HGGs. The analysis was conducted using the “enter” input model, which involved entering all variables into the model in a single step. A conventional *p*-value of less than 0.05 was used to determine the corresponding independent variables that contributed significantly to the differentiation of HGGs from BMs, whereas variables with a *p*-value of more than 0.01 were omitted. In addition, the coefficient of determination (*R*^2^), the *R*^2^-adjusted coefficient, the multiple correlation coefficient (MCC), and the variance inflation factor (VIF, an indicator of multicollinearity) were computed. Following the analysis, the predicted values were saved and then used in a ROC analysis to determine the prediction model’s ability to identify HGGs. Statistical analysis was performed using a commercially available dedicated software, MedCalc version 14.8.1 (MedCalc Software, Mariakerke, Belgium). The workflow diagram is displayed in Figure 2.

## 3. Results

Of the 686 patients who were referred to our department during the study period, 36 were included in this study after applying the inclusion and exclusion criteria. According to their final diagnosis, patients were divided into the HGGs group (*n* = 17; males = 11; females = 6; mean age = 66.3 years; age range = 44–85 years) and BMs group (*n* = 19; males = 10; females = 9; mean age = 54.4 years; age range = 46–83 years).

Seventeen unique texture parameters were provided by the two reduction methods (Fisher and Mutual Information (MI)). Three parameters (10th Percentile (Perc10), Wavelet Energy (WavEnHH_s-2) and Grey Level Non-Uniformity (RZD5GLevNonU)) were selected by both methods. Selected parameters along with the univariate analysis results are displayed in Table 2.

Three parameters (CH5D4DifVarnc, CZ2D4DifVarnc, and WavEnHL_s-3) showed the same results in the ROC analysis, being able to identify primary tumors with at the same rate (Se, 96.67%; Sp, 69.23%). The ROC analysis results are displayed in Table 3 and Figure 3.

The multiple regression analysis showed a coefficient of determination of 0.7153, an *R*^2^-adjusted of 0.6583, and an MCC of 0.8457. No parameters were excluded from the prediction model due to a high VIF. Three parameters (CN6D4Contrast, Perc10, and RZD5GLevNonU) were independent predictors of HGGs (Table 4). The ROC analysis of the predicted values that investigated the diagnostic value of all the parameters combined resulted in a significance level (*p*) of <0.0001, an AUC of 0.992 (95% CI, 0.903–1), and a Youden index of 0.93. Using an associated criterion of >0.57, the prediction model was able to identify HGGs with a sensitivity of 93.33% (95% CI, 77.9–99.2%) and a specificity of 100% (95% CI, 75.3–100%) (Figure 4).

## 4. Discussion

Our results show that seven parameters obtained statistically significant results in the univariate analysis. These parameters were derived from the histogram analysis (Perc10), wavelet decomposition (WavEnHH_s_2 and WavEnHL_s-3), co-occurrence matrix (CN6D4Contrast, CH5D4DifVarnc, and CZ2D4DifVarnc), and the run-length matrix (RZD5GLevNonU).

The 10th percentile (Perc10) was able to identify primary tumors with 81% Se and 85.71% Sp. The percentile index (n) is the point at which n% of the pixel values that form the histogram are deviated to the left [20]. This signifies that 10% of the pixels within the PZ were distributed under higher values for HGGs than for BMs.

Wavelet transformation is a multiresolution technique that aims to transform images into a representation that contains both spatial and frequency information [21]. Wavelet energy quantifies the distribution of energy along the frequency axis over scale and orientation. Energy measures the local uniformity within an image [22]. We obtained higher values for BMs of both wavelet energy parameters (WavEnHH_s-2 and WavEnHL_s-3) that were selected by the reduction techniques. An example of the wavelet multi-level decomposition of a CT image of a patient with glioblastoma is displayed in Figure 5.

The contrast parameter is a measure of the local variations present in an image. If there is a large amount of variation in an image, the contrast will be high [23]. We obtained higher values for the CN6D4Contrast parameter in the peritumoral zone of HGGs, indicating a high rate of pixel intensity variations.

The gray level non-uniformity parameter represents the variability of gray-level intensity values in the image, with a lower value indicating more homogeneity in intensity values [24]. The peritumoral zone of BMs showed lower values of the RZD5GLevNonU parameter than the one of HGGs.

Difference variance is a measure of heterogeneity that places higher weights on differing intensity level pairs that deviate more from the mean [24]. The difference of variance measures the variance of the difference of grey-level values (reflecting the randomness within an image) [25,26]. Both variations of this parameter (CH5D4DifVarnc and CZ2D4DifVarnc) showed higher values for HGGs than for BMs. The parameters’ distribution in CT images of selected cases of HGGs and BMs is shown in Figure 6.

Overall, our results show that the peritumoral zone of HGGs expressed a higher density of pixel intensity, higher rate of pixel variations, lower homogeneity, and higher randomness than peritumoral edema of BMs. These changes may be attributed to the tumoral infiltration of the HGGs’ PZ, which raised the local cellularity (which consequently raised the pixel attenuation) and created a relative inhomogeneity of the peritumoral region. These observations were in accordance with previous studies that we conducted [27,28]. In one study [27], we demonstrated that the apparent diffusion coefficient measured in the peritumoral edema can help to distinguish between the two histopathological entities with 95% Se and 84% Sp. In a subsequent study [28], we demonstrated that the texture parameters extracted from the brain tumors’ PZ on T2-weighted images (T2W) can also successfully distinguish between HGGs and BMs with 100% Se and 66.7% Sp. The Perc10 (along with other histogram parameters) were also selected by the reduction methods. In this study [28], the histogram parameters showed higher values for BMs. The peritumoral zone of BMs seems to exhibit higher signal intensities on T2W images, probably due to the absence of contamination with tumoral cells. The wavelet energy parameters were also selected, again with lower values for HGGs than for BMs. The inhomogeneity of HGGs’ peritumoral zone was also demonstrated through multiple parameters extracted from the run-length matrix, especially short an long run emphasis (RNS6ShrtREmp and RNS6LngREmph) [28] that were not highlighted by the reduction methods in the current study.

Based on magnetic resonance (MRI) images, other studies investigated the utility of radiomic features extracted from the PZ in distinguishing between the two entities. A research study conducted by Skogen et al. [29] showed that the entropy and standard deviation of pixel intensity were able to identify gliomas with a Se of 80% and Sp of 90%. The authors [29] considered these parameters a reflection of the inhomogeneity exhibited by HGGs’ PZ, an observation that is in accordance with our current and previous [28] results. Recently, a multiparametric MR-based RadioFusionOmics model [30] using combined texture parameters extracted from the tumoral and peritumoral zone was able to differentiate the two lesion categories with 85.5% accuracy, 85.6% Se, and 85.3% Sp. Interestingly, none of the above-mentioned studies extracted shape-based textural features in an attempt to differentiate between the two entities. As reported in a study conducted by Della Pepa et al. [31], gliomas’ shape features (regular versus irregular) have a significant role in assessing the resection margins.

Our results demonstrate that TA of the peritumoral zone can successfully distinguish between the two entities on both CT and MRI examinations. Other previously published TA studies that investigated gliomas were based on MRI examinations and mainly focused on the solid tumor, and most often focused on the texture differentiation of high- from low-grade tumors [16,17,32,33].

The conventional imaging differentiation between the two entities relies solely on features related to the morphology, location, and the number of lesions. The multiplicity of the tumoral lesions is considered the most valuable characteristic in distinguishing BMs from gliomas [34]. However, solitary BMs have been reported between a quarter up to half of their encounters [34,35]. Considering that some HGGs can be multicentric, the number criterion may be insufficient for a confident diagnosis between the two. By assessing other morphological features (such as the ratio of the maximal diameter of the peritumoral area to the maximal diameter of the enhancing mass and the aspect of the adjacent edema), the two entities can be distinguished with variable diagnostic rates (up to 45% Se and 44% Sp) [36]. Therefore, there is an obvious need for a more confident diagnostic criterion for distinguishing the two lesions, which could be offered by radiomics through computer-aided diagnosis techniques, as current and previous studies demonstrated good statistical results regarding this approach.

Our choice for relying on CT instead of MRI images also needs to be addressed. First, although now widely accepted, initially there was surprisingly little evidence in the literature that MRI is superior to CT in the detection and characterization of BMs [35]. Second, CT is usually the first imaging modality used in patients with the sudden offset of neurological symptoms, and therefore BMs can be frequently encountered in this type of examination. Third, many institutions employ CT as the initial method of choice in assessing BMs since there is no evidence that MRI-based screening improves clinical outcomes [37]. Finally, to our best knowledge, this is the first study that used TA to differentiate the two entities.

In order to prime their subsequent outgrowth, metastatic cells need to acquire essential traits from the brain’s microenvironment. In this regard, tumor cells’ gene expression undergoes a series of changes that facilitate their adaptation to the new tissue. Interestingly, the tumor cells that metastasize to the brain lose their phosphatase and tensin homolog (PTEN) expression, which is an important tumor suppressor. Subsequently, this process stimulates the outgrowth of brain metastatic tumor cells through reduced apoptosis and enhanced proliferation [38]. On the other hand, glioblastomas‘ microenvironment is highly variable, and it consists of extracellular matrix components, soluble factors, and tissue-resident cell types, together with the resident or recruited immune cells. The latter form the immune microenvironment, which is responsible for a substantial part of the tumoral volume [39]. Therefore, it is possible that the CT features can also reflect the different tumoral microenvironments.

Our study has several limitations. There was no direct correlation between the brain region comprised within the ROI and the microscopic findings due to the retrospective nature of our study, which did not allow direct coordination between the surgical sampling and the pathological analysis. In addition, owing to its retrospective design, the study may present selection bias. The study group was relatively small due to the protocols of our healthcare institution and to the strict inclusion and exclusion criteria. Additionally, the ROI definition comprised a single cross section and not a three-dimensional (3D) volume analysis that could have comprised more texture information. However, the 3D analysis would be hard to adopt in clinical practice because it requires long segmentation times and most likely increases operator variability. The fact that the inter- or intra-reader agreement was not assessed also constitutes a limitation. However, previous studies following the same method stated that inter/intraobserver variability assessment is not mandatory with semi-automatic ROI positioning because it has low variability rates [40]. Additionally, the software used in this study (MaZda, Institute of Electronics, Technical University of Lodz, Lodz, Poland) may be viewed as outdated. However, it may be the only program that provides built-in reduction and classification methods in an intuitive front-end that can be used even by nonimage processing specialists, such as regular physicians.

## 5. Conclusions

Radiomic analysis of brain tumors’ peripheral zone can successfully distinguish between gliomas and solitary brain metastases. The texture parameters may reflect the microscopic inhomogeneity produced by the neoplastic infiltration in the adjacent edema of primary malignancies, although this premise needs further validation through a study that implies direct coordination between the CT and the pathological analysis.

## Figures and Tables

**Figure 1 brainsci-12-00109-f001:**
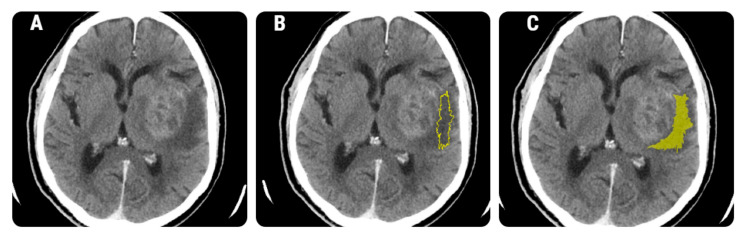
The definition of a region of interest (ROI) in the peritumoral region. (**A**) Axial contrast-enhanced computed tomography (CT) scan of a 68-year-old patient with histologically proven glioblastoma; (**B**) the ROI (yellow line) that was automatically delineated by the software based on geometry and gradient coordinates; (**C**) the final ROI after the manual corrections were applied.

**Figure 2 brainsci-12-00109-f002:**
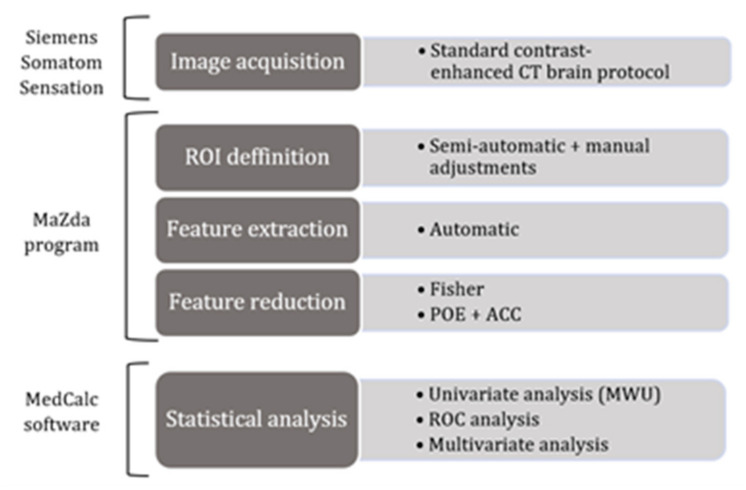
Image processing workflow diagram. CT, computed tomography; ROI, region of interest; POE, probability of classification error; ACC, average correlation coefficients; MWU, the Mann–Whitney U test; ROC, receiver operating characteristics.

**Figure 3 brainsci-12-00109-f003:**
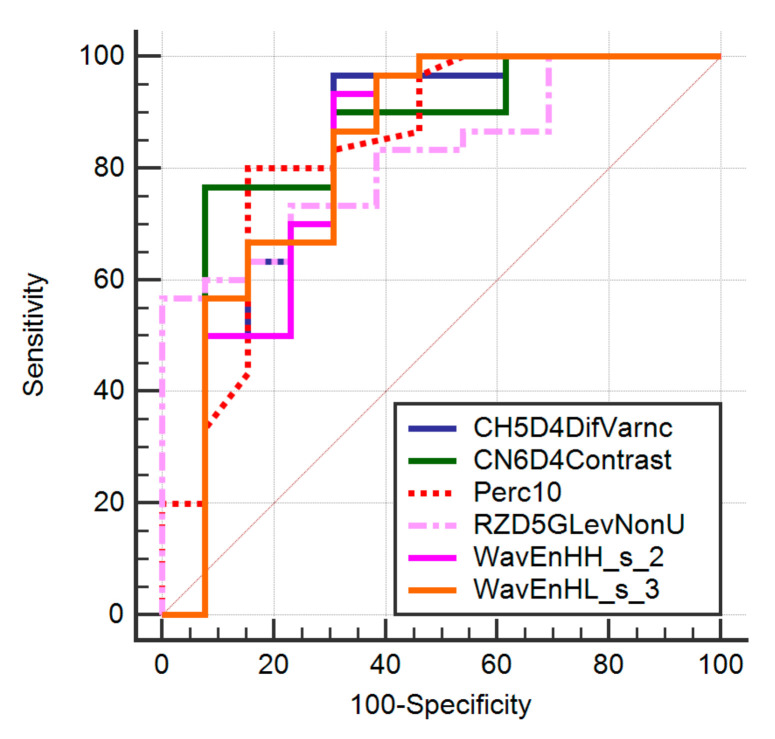
Receiver operating characteristics analysis curve showing the diagnostic utility of texture parameters in differentiating gliomas from brain metastases.

**Figure 4 brainsci-12-00109-f004:**
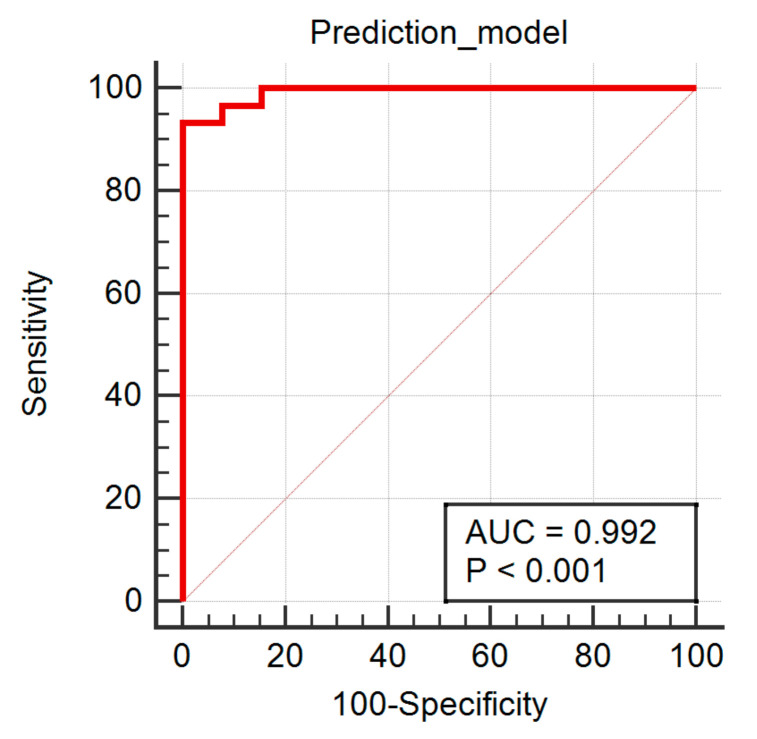
Receiver operating characteristics curve of the prediction model in the diagnosis of high-grade gliomas. AUC, area under the curve; *p*, statistical significance level.

**Figure 5 brainsci-12-00109-f005:**
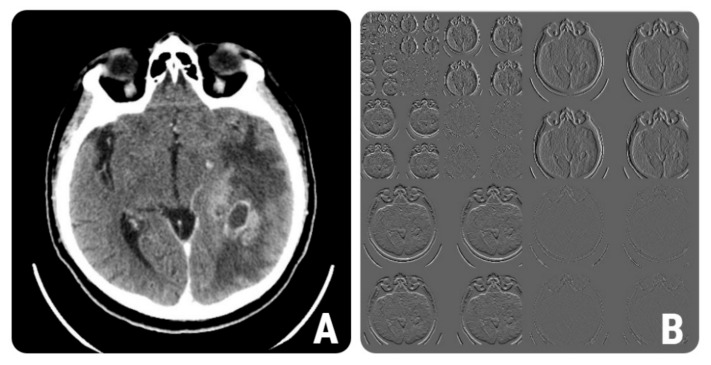
(**A**) Contrast-enhanced CT image of a 57-year-old patient with histologically proven grade IV glioblastoma. (**B**) The wavelet multi-step decomposition of image (**A**).

**Figure 6 brainsci-12-00109-f006:**
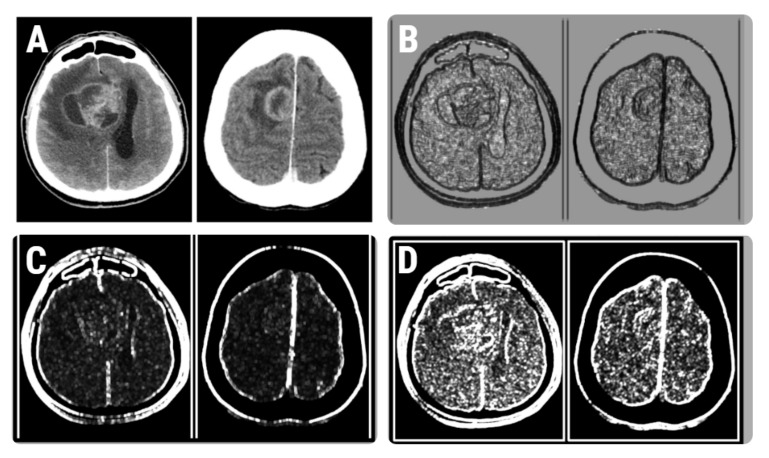
(**A**) CT images of patients with histologically proven glioblastoma (left) and brain metastases (right). (**B**–**D**) Texture maps that show the distribution of the RZD5GLevNonU (**B**), CH5D4DifVarnc (**C**), and contrast (**D**) parameters in the selected CT images.

**Table 1 brainsci-12-00109-t001:** Texture parameters.

Class	Texture Features	Computation Parameters	Variations
Run-length matrix (*n* = 20)	RLNonUni, GLevNonU, LngREmph, ShrtREmp, Fraction	6 bits/pixel	4 directions
Wavelet transformation (*n* = 20)	WavEn	5 scales	4 frequency bands
Co-occurrence matrix (*n* = 220)	AngScMom, Contrast, Correlat, SumOfSqs, InvDfMom, SumAverg, SumVarnc, SumEntrp, Entropy, DifVarnc, DifEntrp	6 bits/pixel; 5 between-pixels distances	4 directions
Histogram (*n* = 5)	Mean, Variance, Skewness, Kurtosis, Perc.01–99%	-	-
Absolute gradient (*n* = 5)	GrMean, GrVariance, GrSkewness, GrKurtosis, GrNonZeros	4 bits/pixel	-
Auto-regressive model	Teta 1–4, Sigma	-	-

*n* = total number of parameters computed from each class; RLNonUni, run-length nonuniformity; GLevNonU, grey level nonuniformity; LngREmph, long-run emphasis; ShrtREmp, short-run emphasis; Fraction, the fraction of image in runs; WavEn, wavelet energy; AngScMom, angular second moment; Correlat, correlation; SumOfSqs, the sum of squares; InvDfMom, inverse difference moment; SumAverg, sum average; SumVarnc, sum variance; SumEntrp, sum entropy; DifVarnc, difference variance; DifEntrp, difference entropy; Mean, histogram’s mean; Variance, histogram’s variance; Skewness, histogram’s skewness; Kurtosis, histogram’s kurtosis; Perc.01–99%, 1–99% percentile; GrMean, absolute gradient mean; GrVariance, absolute gradient variance; GrSkewness, absolute gradient skewness; GrKurtosis, absolute gradient kurtosis; GrNonZeros, percentage of pixels with nonzero gradient; Teta 1–4, parameters θ1–θ4; Sigma, parameter *σ*.

**Table 2 brainsci-12-00109-t002:** Parameters selected by the two reduction techniques and the univariate analysis (Mann–Whitney U test) results.

Texture Parameter	*p*-Value	Primary Tumors		Metastases	
		Median	IQR	Median	IQR
Fisher					
Perc10	**<0.001**	32.8	24–38	8.12	6–14
WavEnHH_s-2	**0.0013**	8.5	3.95–10.87	15.8	11–20.1
CN6D4Contrast	**<0.001**	32. 15	24.3–37.8	18.6	8.6–22.14
Teta3	0.6	0.17	0.01–0.41	0.19	0.13–0.61
Kurtosis	0.33	10.6	0.13–68.4	18.8	28.2–59.3
CN6D5Correlat	0.06	0.58	0.51–0.77	0.51	0.26–0.64
RZD5GLevNonU	**<0.001**	3041.8	1310.7–3969.2	1081.2	641.01–1922.92
RZD3Fraction	0.041	0.77	0.7–0.81	0.68	0.41–0.77
CH5D4DifVarnc	**<0.001**	20.43	12.51–24.8	6.23	3.3–15.6
Perc50	0.07	19.24	11–26	16.43	7–25
POE+ACC					
CZ2D4DifVarnc	**<0.001**	22.13	12.94–26.11	7.26	3.81–15.41
WavEnHL_s-3	**<0.001**	10.65	5.33–21.12	28.68	16.2–38.02
CV3S6SumAverg	0.049	64.15	39.12–84.9	52.8	26.7–74.17
RVD6LngREmph	0.62	2.31	1.81–3.19	5.73	2.46–38.14
CZ5S6Correlat	0.01	0.56	0.21–0.82	0.29	0.01–0.65
CN4S6Entropy	0.03	1.13	0.04–2.27	3.01	1.7–5.89
CV1S6AngScMom	0.46	0.12	0.01–0.22	0.29	0.06–0.36

Bold values are statistically significant. POE + ACC, probability of classification error and average correlation coefficients.

**Table 3 brainsci-12-00109-t003:** The receiver operating characteristic analysis results of texture parameters in the diagnosis of primary tumors.

Texture Parameter	AUC	Sign.lvl.	Youden Index	Cut-Off	Se (%)	Sp (%)
Perc10	0.84 (0.7–0.9)	**<0.0001**	0.66	>21	81 (62.3–91.2)	85.71 (56.2–97.61)
WavEnHH_s-2	0.81 (0.6–0.91)	**0.0004**	0.6256	≤14.17	93.33 (77.9–99.2)	69.23 (38.6–90.9)
CN6D4Contrast	0.84 (0.65–0.91)	**<0.0001**	0.67	>22.26	77.8 (58.3–91.2)	93.22 (65.7–98.7)
RZD5GLevNonU	0.82 (0.67–0.92)	**<0.0001**	0.56	>2447.78	56.67 (37.4–74.5)	100 (75.3–100)
CH5D4DifVarnc	0.82 (0.67–0.92)	**0.0002**	0.65	>17.69	96.67 (82.8–99.9)	69.23 (38.6–90.9)
CZ2D4DifVarnc	0.82 (0.67–0.92)	**0.0001**	0.66	>21.05	96.67 (82.8–99.9)	69.23 (38.6–90.9)
WavEnHL_s-3	0.82 (0.67–0.92)	**0.0001**	0.58	≤27.2	96.67 (82.8–99.9)	69.23 (38.6–90.9)

The values corresponding to 95% confidence intervals are shown in parentheses. Bold values are statistically significant. AUC, area under curve; Sign.lvl., significance level; Se, sensitivity; Sp, specificity.

**Table 4 brainsci-12-00109-t004:** Multivariate analysis results showing the texture parameters independently associated with the presence of high-grade gliomas.

Independent Variables	Coefficient	Std. Error	*p*	r_partial_	r_semipartial_	VIF
CH5D4DifVarnc	0.05461	0.04878	0.2705	0.1859	0.101	119.563
CN6D4Contrast	−0.0292	0.009469	**0.004**	−0.4623	0.2782	7.503
CZ2D4DifVarnc	−0.02923	0.04013	0.4713	−0.1222	0.06569	84.372
Perc10	0.0194	0.003637	**<0.0001**	0.6696	0.481	1.747
RZD5GLevNonU	0.00008993	3.28E-05	**0.0096**	0.4203	0.2472	1.223
WavEnHH_s_2	−0.01056	0.02459	0.6702	−0.07241	0.03874	12.831
WavEnHL_s_3	0.0004019	0.01425	0.9777	0.004767	0.002544	18.931

Bold values are statistically significant. Std. Error, standard error; r_partial_, partial correlation; r_semipartial_, semipartial correlation; *p*, multivariate analysis result; VIF, Variance Inflation Factor.

## Data Availability

The data presented in this study are available on request from the corresponding author.

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
