# Peer review of "CT in the Differentiation of Gliomas from Brain Metastases: The Radiomics Analysis of the Peritumoral Zone"

_brainsci, 2022, doi:10.3390/brainsci12010109_

Round 1
Reviewer 1 Report
Interesting and well written study regarding the differential diagnosis between highgrade gliomas and cerebral metastasis using the radiomics of the peritumoral zone with ct.
few comments/points to be corrected:
- in the abstract I would suggest to better delineate your conclusions.
- In the introduction you should better explain what is radiomics and what are the innovations given by radiomics, to let readers become more familiar with this.
- better explain why the study is conducted with ct, since with MRI you can easily differentiate HGG to BM.
- Usually the differential dignosis between HGG and BM is possibile, just in a minority of cases this is difficult. Explain in this context what is the importance of your study.
- in discussion add some comments regarding differential diagnosis of these lesions with MRI, which are the differences compared to your work?
- In conclusion better delineate the major findings of your work
- Minor english grammar corrections.
Reviewer 2 Report
This topic is very topical and often represents a challenge for neuroradiologists, please look at these points to improve:
- Line 45: "The peritumoral zone (PZ) of the two entities has specific microscopic characteristics" What do authors mean for peritumoral zone? Explain better.
- Lines 39-41: "The differentiation between high-grade gliomas (HGGs) and solitary brain metastases (BMs) is crucial, as they imply separate clinical and surgical management strategies [1]." But also different outcome and overall survival.
- Lines 56-57: "This is the first study that aimed to extract the texture information from the PZ of solitary brain tumors based on CT images" Are you sure about it? Maybe you can add: "To the best of our knowledge, this is the first... "
- In methods, please state within which perimeter is the peritumor area considered.
- Figure legend n° 3 should be improved.
- Lines 227-235: Although this manuscript is a more radiological type paper, a small report about histological molecular component must be made in the discussion, as well as the importance of the microenvironment in HGG and metastases.
- In the conclusion section, please add some results of your research. What this paper add new to the literature?
- Statistical analysis is very good.
Overall a good paper, but some revisions are needed.
Round 2
Reviewer 2 Report
Authors solved all my criticisms.